# Identification of Novel Gene Cluster Potentially Associated with Insecticide Resistance in *Anopheles gambiae* s.l.

**DOI:** 10.3390/genes16091018

**Published:** 2025-08-28

**Authors:** Hyacinthe Dipina Ki, Mahamadi Kientega, Sabéré O. G. Yemien, Hamidou Maiga, Nouhoun Traoré, Koama Bayili, Moussa Namountougou, Abdoulaye Diabaté

**Affiliations:** 1Institut de Recherche en Sciences de la Santé (IRSS), Bobo-Dioulasso 01 BP 545, Burkina Faso; mkient54@gmail.com (M.K.); gillesyemien@gmail.com (S.O.G.Y.); maigahamid@yahoo.fr (H.M.); nouhoun89@gmail.com (N.T.); kwamajacques@yahoo.fr (K.B.); 2Unité de Formation et de Recherche en Sciences de la Vie et de la Terre, Université Nazi Boni, Bobo-Dioulasso 01 BP 1091, Burkina Faso; namountougou_d@yahoo.fr

**Keywords:** malaria, *An. gambiae* s.l., insecticide resistance, aldehyde oxidase, genomic surveillance

## Abstract

Background/Objectives: Despite the increasing emergence of resistance, insecticide-based tools remain the primary method for malaria vector control in Africa. To maintain the effectiveness of these interventions, continuous monitoring and identification of novel resistance mechanisms is essential. This study aimed to investigate potential new insecticide resistance genes in the *Anopheles gambiae* complex. Methods: We analyzed whole-genome sequencing data from the *An. gambiae* 1000 Genomes Project. A broad range of genomic analysis techniques and tools were used to identify and explore genetic variation in the candidate resistance genes. Results: High haplotype homozygosity values, indicative of positive selection, were detected in a 2L chromosomal region corresponding to an aldehyde oxidase gene cluster (*AGAP006220*, *AGAP006221*, *AGAP006224*, *AGAP006225*, *AGAP006226*). Single nucleotide polymorphisms (SNPs) have been identified in these genes with frequencies up to 100%, including 569, 691, 1433, 978, and 811 non-synonymous SNPs in *AGAP006220*, *AGAP006221*, *AGAP006224*, *AGAP006225*, and *AGAP006226*, respectively. Copy number variations (CNVs) such as deletions and amplifications were also identified at low frequencies (<12%). Population structure analyses revealed adaptive and geographic gene flow between *An. gambiae* and *An. coluzzii*. Conclusions: This study provides evidence that aldehyde oxidase genes may contribute to insecticide resistance in *An. gambiae* s.l. populations. These results highlight the importance of genomic surveillance for detecting novel resistance loci and guiding the development of improved vector control strategies under changing ecological and evolutionary conditions.

## 1. Introduction

Malaria remains one of the leading causes of morbidity and mortality in sub-Saharan Africa, especially among children under five years and pregnant women [1]. Despite recent advances in malaria vaccine development, insecticide-based interventions, including long-lasting insecticidal nets (LLINs) and indoor residual spraying (IRS), remain the most effective prevention methods [2]. The success of these interventions depends on a thorough understanding of vector population dynamics, especially their susceptibility to insecticides, to guide the optimal deployment of control tools. In mosquitoes, two major mechanisms drive insecticide resistance: target-site and metabolic resistance [3,4,5]. Target-site resistance involves genetic changes that alter the protein targeted by the insecticide, whereas metabolic resistance results from changes in the sequence or expression of detoxification enzymes and pathways that metabolize or sequester the insecticide [6]. In addition to these mechanisms, behavioral and cuticular resistance have also been described in malaria vectors [7]. The rise in resistance remains a major concern for malaria control and calls for continued resistance surveillance and rapid development of alternative control tools to accelerate malaria elimination.

Although target-site resistance has been extensively described in the African malaria vector *Anopheles gambiae* s.l., metabolic resistance remains poorly characterized [8], likely due to the number of gene families involved in detoxification, redundancy among their members, and the multiple mechanisms by which metabolic resistance can arise [9]. However, research on this mechanism is becoming increasingly critical, as recent studies have shown a growing contribution of detoxification genes to the resistance phenotype in *An. gambiae* s.l. populations [10,11,12]. This trend is likely driven by novel mutations in genes encoding detoxifying enzymes that are associated with resistance phenotypes in members of the *An. gambiae* complex [13]. Key detoxifying enzymes include cytochrome P450 monooxygenases, glutathione S-transferases, and carboxylesterases [14].

A central challenge in malaria vector control is the identification of the genomic basis of resistance in ways that can inform field-based control programs [15]. Traditionally, insecticide resistance has been monitored using phenotypic assays such as WHO tube tests or CDC bottle bioassays. While useful, these methods are labor-intensive, time-consuming, and require substantial logistical resources. Advances in genomics now provide powerful alternatives, enabling the detection and surveillance of molecular markers associated with resistance in natural mosquito populations [16]. To date, most genomic studies have focused on regions of the genome already linked to resistance. However, given the rapid and dynamic evolution of insecticide resistance, including the emergence of novel mechanisms, whole-genome analysis offers an optimal approach. Genome-wide scans for signatures of adaptive evolution, such as selective sweeps, provide a comprehensive framework for detecting resistance loci beyond known targets.

In this study, we investigated genetic variation within the genomic region containing aldehyde oxidase genes and its potential implications for insecticide resistance in *An. gambiae* s.l. populations.

## 2. Materials and Methods

### 2.1. Data Collection and Analysis

The genomic data analyzed in this study were obtained from the *An. gambiae* 1000 Genomes Project, phase 3 (Ag1000G; dataset Ag3.0), released in February 2021 [17]. Mosquitoes representing three *Anopheles* species, *An. gambiae*, *An. coluzzii*, and *An. arabiensis*, were collected between 2004 and 2015 in 11 countries across sub-Saharan Africa (Figure 1, Appendix A). Detailed protocols for mosquito sampling, specimen storage, and genomic data management, including access rights, are available on the MalariaGEN website [18].

Library preparation and sequencing were performed at the Wellcome Sanger Institute. All specimens were sequenced to a target coverage of 30× using the Illumina HiSeq 2000 and the Illumina HiSeq X platforms [18].

Alignment, SNP calling, and sample quality control (QC) were carried out by the MalariaGEN Resource Centre Team. Sequencing reads were aligned to the AgamP4 reference genome using BWA version 0.7.15. Indel realignment was performed with the GATK version 3.7-0 RealignerTargetCreator and IndelRealigner, and SNPs were called using the GATK version 3.7-0 UnifiedGenotyper. Genotypes were called for each sample independently, considering all possible alleles at genomic sites where the reference base was not “N.” Following variant calling, both samples and variants underwent multiple QC analyses to ensure data integrity and accuracy [18].

### 2.2. Detection of Genome Regions Under Recent Positive Selection and Identification of New Candidate Genes

Genome-wide scans for selection (GWSS) were performed on *An. gambiae* s.l. populations to detect signals of recent selection using the Garud H12 statistic [16], a metric particularly sensitive to recent selective sweeps, such as those driven by insecticide pressure. Analyses were first performed at the country level and then stratified by both country and species. For each analysis, the window size, defined by a specific number of SNPs, was calibrated to balance resolution and signal detection. Genomic regions showing signatures of selective sweeps were screened for insecticide resistance genes. Potential novel candidates were identified through literature searches using resources including VectorBase, NCBI, and Google Scholar.

### 2.3. Genetic Variability

For each aldehyde oxidase gene, the corresponding transcript was determined. Single nucleotide polymorphisms (SNPs) were identified and categorized according to their predicted functional effects. SNP frequencies within populations were calculated, with further filtering to retain only high-frequency non-synonymous SNPs. Copy number variation (CNV) was assessed by computing the frequencies of gene amplifications and deletions in the aldehyde oxidase gene regions across populations.

Haplotype networks were constructed for aldehyde oxidase genes to investigate potential gene flow between countries and species. Haplotypes carrying the predominant high-frequency mutations were analyzed using a maximum genetic distance threshold of two SNPs.

Population structure was assessed using principal component analysis (PCA) and pairwise *Fst* estimates. Genetic diversity summary statistics were calculated from SNPs located within the aldehyde oxidase gene regions.

## 3. Results

### 3.1. Signals of Positive Selection–Identification of Potential New Insecticide Resistance Genes

We analyzed a total of 1908 individual genomes from *An*. *gambiae* s.s., *An*. *coluzzii*, and *An. arabiensis*. Genome-wide scans for selection were performed using the Garud H12 statistic in windows ranging from 1000 to 2000 SNPs, depending on the population. Several peaks of H12 values were detected across different chromosome arms. These peaks correspond to genomic regions containing the Vgsc, Ace1, Rdl, and GSTe3 genes, indicating selective sweeps likely driven by positive selection, as previously reported in *An. gambiae* populations (Figure 2).

In addition to these well-characterized resistance loci, a strong H12 peak was observed at a novel locus on chromosome 2L (28,510,000–28,590,000), indicating a recent selective sweep in this region. This genomic region corresponds to a cluster of genes encoding detoxification enzymes, including five aldehyde oxidases (*AGAP006220*, *AGAP006221*, *AGAP006224*, *AGAP006225*, *AGAP006226*), two glucosyl/glucuronosyl transferases (*AGAP006222*, *AGAP006223*), and two carboxylesterases (*AGAP006227*, *AGAP006228*). Population-specific analyses revealed that in *An*. *gambiae* s.s., the sweep occurred in Burkina Faso, Ghana, and Guinea. In *An*. *arabiensis*, it was detected in Tanzania and Malawi, though at lower frequencies. In contrast, the sweep was entirely absent in *An*. *coluzzii* (Figure 2).

We focused subsequent analyses on the aldehyde oxidases, as three of these genes were located directly under the peak of the selective sweep and, based on the literature evidence, appear to be strong candidates for involvement in insecticide resistance.

### 3.2. SNPs in the Aldehyde Oxidase Genes

Single nucleotide polymorphism (SNP) variation was analyzed in aldehyde oxidase genes across the five countries, Burkina Faso, Ghana, Guinea, Tanzania, and Malawi, where these genes showed evidence of positive selection. All five genes showed a high level of polymorphism across the different populations. Specifically, we identified 2202 SNPs (including 569 non-synonymous coding SNPs) in *AGAP006220*, 2661 SNPs (691 non-synonymous) in *AGAP006221*, 2698 SNPs (1433 non-synonymous) in *AGAP006224*, 2515 SNPs (978 non-synonymous) in *AGAP006225*, and 2104 SNPs (811 non-synonymous) in *AGAP006226*. The analyses identified a substantial number of non-synonymous coding SNPs at high frequencies across all five genes and in all three species, with some reaching 100% frequency in several cohorts. To focus on the most prevalent variants, the dataset was refined to include only those SNPs with minimum frequencies of 40, 50, or 75% in at least one sample cohort. The refinement showed 21 SNPs at frequencies higher than 40% in *AGAP006220* and *AGAP006225*, and 56 SNPs above 75% in *AGAP006221*, *AGAP006224*, and *AGAP006226* (Figure 3 and Appendix A). These patterns indicate that these high-frequency variants may be subject to strong selective pressure and may play a significant role in the adaptation of malaria vectors to environmental challenges, including insecticide exposure.

### 3.3. CNVs in the Aldehyde Oxidase Genes

Copy number variation (CNV) frequencies were assessed for the five aldehyde oxidase genes across populations from Burkina Faso, Ghana, Guinea, Tanzania, and Malawi. The analysis revealed the presence of CNVs in all five genes. Deletions were detected only in *AGAP006224*, while amplifications were observed in all five genes. Overall, amplification frequencies were low (≤3%) across most of the genes and populations, except for *An. arabiensis* from Tanzania, where *AGAP006225* and *AGAP006226* exhibited amplification rates of 10.59% and 11.76%, respectively (Figure 4).

### 3.4. Gene Flow, Population Structure, and Genetic Diversity at the Aldehyde Oxidases Locus

To investigate gene flow at the aldehyde oxidases locus, the haplotype network technique was used to visualize the relationships between haplotypes from different species and countries and to show how variants are shared between populations. The results revealed strong signals of adaptive gene flow within *An. gambiae* s.l. populations in West Africa, as well as within populations in East Africa. In West Africa, the largest haplotype network comprised 163 identical haplotypes shared across all vector populations. A similar pattern was observed in East Africa, where 52 identical haplotypes were shared between *An. gambiae* s.l. populations from Tanzania and Uganda (Figure 5). Evidence of interspecific gene flow was also apparent, with a large haplotype cluster of 1001 identical sequences shared between *An. gambiae* (n = 609) and *An. coluzzii* (n = 392), indicating strong adaptive gene flow between these two species. In contrast, no shared haplotypes were detected between *An. arabiensis* and either *An. gambiae* or *An. coluzzii* (Figure 6).

These patterns were corroborated by population structure and genetic diversity analyses performed at the aldehyde oxidases locus across the 11 countries. Genetic differentiation (*F_ST_*) between *An. arabiensis* and the two other species were generally higher than those observed between *An. gambiae* and *An. coluzzii*, indicating greater genetic divergence (Appendix A). Diversity indices also supported these trends, although some within-species variation was observed. Both measures of diversity (θπ and θw) showed similar results for *An. gambiae* and *An. coluzzii* compared to *An. arabiensis* (Figure 7). Principal component analysis (PCA) showed a distinct and homogeneous cluster of *An. arabiensis*, whereas *An. gambiae* and *An. coluzzii* clustered together, further supporting genetic exchange between these two species (Figure 8). Overall, these findings suggest that *An. gambiae* and *An. coluzzii* share a similar demographic history.

## 4. Discussion

Malaria control remains a major challenge in sub-Saharan Africa. Despite ongoing interventions, the region still experiences the highest global burden of the disease due to complicating factors such as insecticide resistance, changing vector behavior, and rapid urbanization [2,19]. Strengthening surveillance, particularly through genomic tools, offers new opportunities to enhance the effectiveness of vector control strategies [20]. In particular, identifying novel resistance-associated genes is essential for enabling early detection and monitoring within vector populations. These insights help guide and inform the selection and rotation of insecticides, reducing the spread of resistance variants [21]. Recent advances in genomics offer important opportunities to study the genomic architecture and evolution of malaria vectors, enabling high-resolution mapping of resistance genes, population structure, and adaptive evolution [21,22].

### 4.1. Aldehyde Oxidases Under Positive Selection

Garud H12 statistic revealed evidence of positive selection in aldehyde oxidase genes within *An. gambiae* and *An. arabiensis* populations across sub-Saharan Africa. The absence of such selection in *An. coluzzii* is consistent with recent findings on the carboxylesterases Coeae1f and Coeae2f, which, as previously mentioned, are located on the same locus [23]. Aldehyde oxidases are known for their broad substrate specificity and marked species differences [24].

### 4.2. Potential Role of Aldehyde Oxidases in Insecticide Resistance

Literature review indicates that aldehyde oxidase genes may play an important role in the adaptation of malaria vectors to environmental pressures, including those induced by insecticide-based control measures. Aldehyde oxidase is a cytosolic enzyme that belongs to the family of structurally related molybdo-flavoproteins like xanthine oxidase. These enzymes have originated from gene duplication events of an ancestral eukaryotic xanthine oxidoreductase gene first identified in the flatworm, *Caenorhabditis elegans* [24]. Aldehyde oxidases are common enzymes involved in detoxifying and breaking down many substances across various organisms. Although references to aldehyde oxidase and its role in metabolism date back to the 1950s [24], the significance of this enzyme in insecticide metabolism has only emerged in recent years [25]. Like cytochromes P450, aldehyde oxidases contribute to oxidation by carrying out nucleophilic attacks using oxygen derived from water [26]. In *Culex* mosquitoes, elevated aldehyde oxidase activity has been documented in resistant strains, indicating a potential selective advantage under insecticide exposure [27,28]. In *An. gambiae* s.l., aldehyde oxidases have been found co-amplified with esterases known to confer resistance to the organophosphate pirimiphos-methyl [23,27]. In addition, in vitro assays have demonstrated their ability to metabolize neonicotinoids including imidacloprid, thiamethoxam, clothianidin, and dinotefuran [29], indicating a potential role in resistance to this newer insecticide class. This is particularly relevant given the increasing interest in neonicotinoids as alternative tools for vector control in areas facing widespread multi-class resistance [30].

### 4.3. Genetic Mechanisms: SNPs and CNVs

In addition to positive selection, variant analyses revealed non-synonymous single nucleotide polymorphisms (SNPs) and copy number variations (CNVs), including gene amplifications, in the genomic region encompassing aldehyde oxidase genes. Both SNPs and CNVs are recognized as important drivers of insecticide resistance evolution [31,32,33,34] particularly in metabolic resistance, where they can increase gene expression [3,35]. Given the relatively low CNV frequencies detected in our results, we hypothesize that the putative resistance conferred by these detoxification genes may be more strongly associated with SNPs, which occurred at higher frequencies. This is consistent with previous evidence showing that SNPs are key determinants of variation in aldehyde oxidase expression and protein activity [24].

### 4.4. Gene Flow and Geographic Variation

The adaptive introgression detected between *An. gambiae* and *An. coluzzii* supports previous studies reporting extensive genetic exchange between these closely related species [22]. Although no significant selective sweep was detected in *An. coluzzii*, the ongoing and increasing gene flow suggests that aldehyde oxidase genes could come under selective pressure in this species as well. Based on the observed spatial patterns of gene flow, these putative resistance markers are expected to expand to large areas across the continent. 

### 4.5. Implications for Vector Control

This study highlighted the ongoing evolution of the *An. gambiae* complex genome in response to environmental pressures and vector control interventions. Genomic changes, including mutations and gene duplications, are driven by factors such as exposure to insecticides, environmental changes and ecological shifts. These evolutionary processes can lead to increased resistance and altered behavior, which complicate malaria control efforts. Continuous genomic surveillance is therefore crucial to track and effectively respond to these changes.

## 5. Conclusions

This study highlights the importance of genomic data in malaria vector surveillance, particularly for detecting emerging mechanisms of insecticide resistance. Through whole genome analysis, we identified a previously unreported locus on chromosome 2L comprising detoxification genes encoding aldehyde oxidases (*AGAP006220*, *AGAP006221*, *AGAP006224*, *AGAP006225*, and *AGAP006226*) under recent positive selection. High-frequency non-synonymous SNPs, 569, 691, 1433, 978, and 811 in the respective genes were detected, with some reaching 100% frequency, alongside copy number variations (deletions and amplifications) at low frequencies (<12%). Population structure analysis revealed adaptive and geographic gene flow between *An. gambiae* and *An. coluzzii*, highlighting historical connectivity and potential gene exchange.

While aldehyde oxidases have previously been implicated in insecticide resistance in *Culex* mosquitoes, our findings provide the first genomic evidence supporting their potential involvement in insecticide resistance in *An. gambiae* s.l. populations. Further functional and gene expression studies will be necessary to validate this role. If confirmed, these genes could serve as novel molecular markers in resistance monitoring programs, enhancing the design of integrated and adaptive vector control strategies in Africa.

## Figures and Tables

**Figure 1 genes-16-01018-f001:**
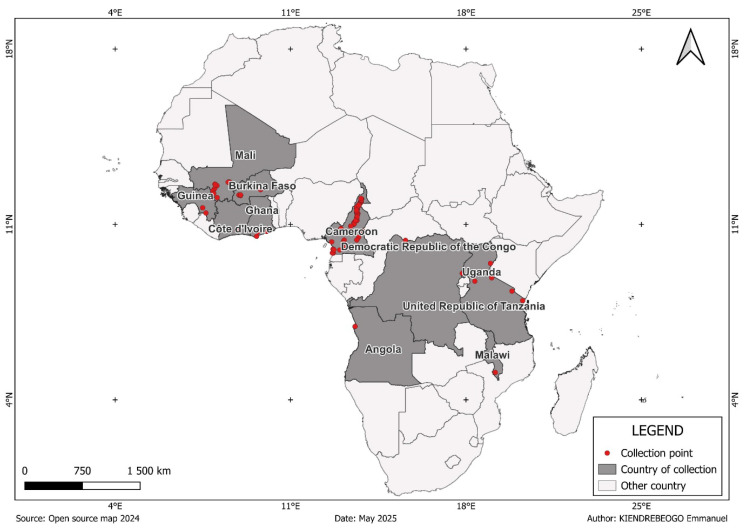
Countries and collection sites.

**Figure 2 genes-16-01018-f002:**
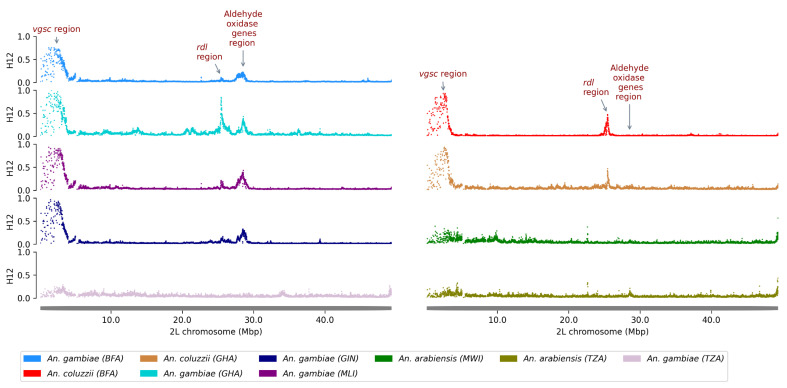
H12 statistic on the 2L chromosomal arm of mosquitoes from Burkina Faso (BFA), Ghana (GHA), Mali (MLI), Guinea (GIN), Malawi (MWI), and Tanzania (TZA). Selective sweeps are suggested by peaks. A cluster of five aldehyde oxidases was found under positive selection in *Anopheles gambiae* and *An. arabiensis*: *AGAP006220* (2L: 28,512,602–28,517,680), *AGAP006221* (2L: 28,518,055–28,523,900), *AGAP006224* (2L: 28,528,758–28,533,199), *AGAP006225* (2L: 28,534,732–28,539,416), and *AGAP006226* (2L: 28,540,651–28,545,294).

**Figure 3 genes-16-01018-f003:**
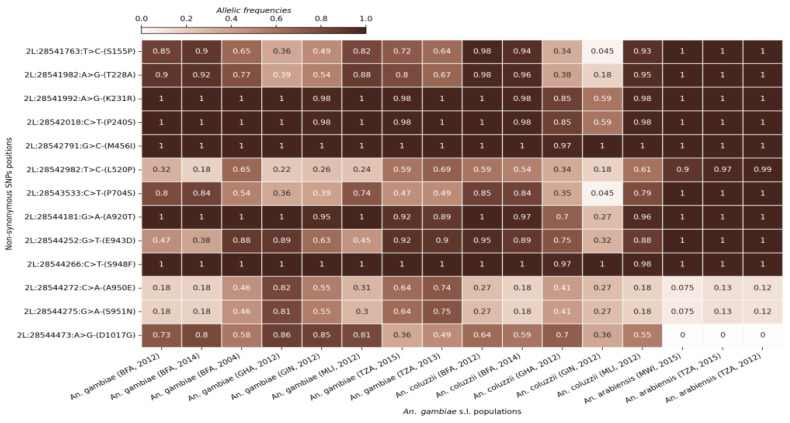
SNP allele frequencies in the *AGAP006226* gene in the different *An*. *gambiae* s.l. populations. The X-axis shows the different cohorts (country, species, and sampling period). The Y-axis shows the SNPs positions in the gene and the corresponding amino acid change. The gradient color bar shows the distribution of the allelic frequencies.

**Figure 4 genes-16-01018-f004:**
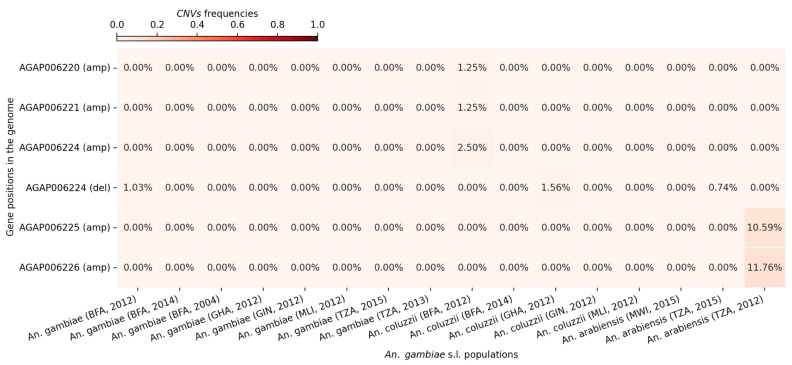
CNV frequencies at the aldehyde oxidases genes in the different *An. gambiae* s.l. populations. The X-axis shows the different cohorts (country, species, and sampling period). The Y-axis shows the genes ID and the CNV type (del: deletion or amp: amplification). The gradient color bar shows the distribution of the CNV frequencies.

**Figure 5 genes-16-01018-f005:**
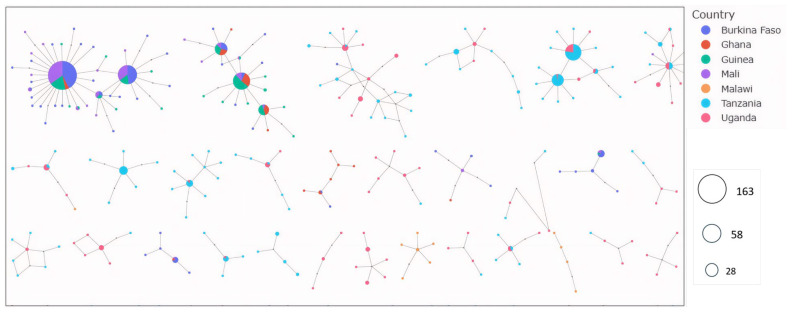
Haplotype networks in five aldehyde oxidases (*AGAP006220*, *AGAP006221*, *AGAP006224*, *AGAP006225*, and *AGAP006226*) showing gene flow between countries. Identical haplotypes are represented by nodes (circles). The larger the node, the greater the number of identical haplotypes. Maximum distance set to two, i.e., each node is separated by a maximum genetic distance of 2 SNPs. Color indicates the country.

**Figure 6 genes-16-01018-f006:**
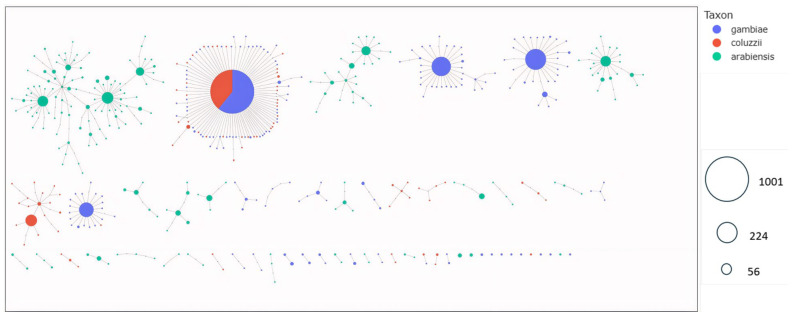
Haplotype networks in five aldehyde oxidases (*AGAP006220*, *AGAP006221*, *AGAP006224*, *AGAP006225*, and *AGAP006226*) showing gene flow between species. Identical haplotypes are represented by nodes (circles). The larger the node, the greater the number of identical haplotypes. Maximum distance set to two, i.e., each node is separated by a maximum genetic distance of 2 SNPs. Color indicates the species.

**Figure 7 genes-16-01018-f007:**
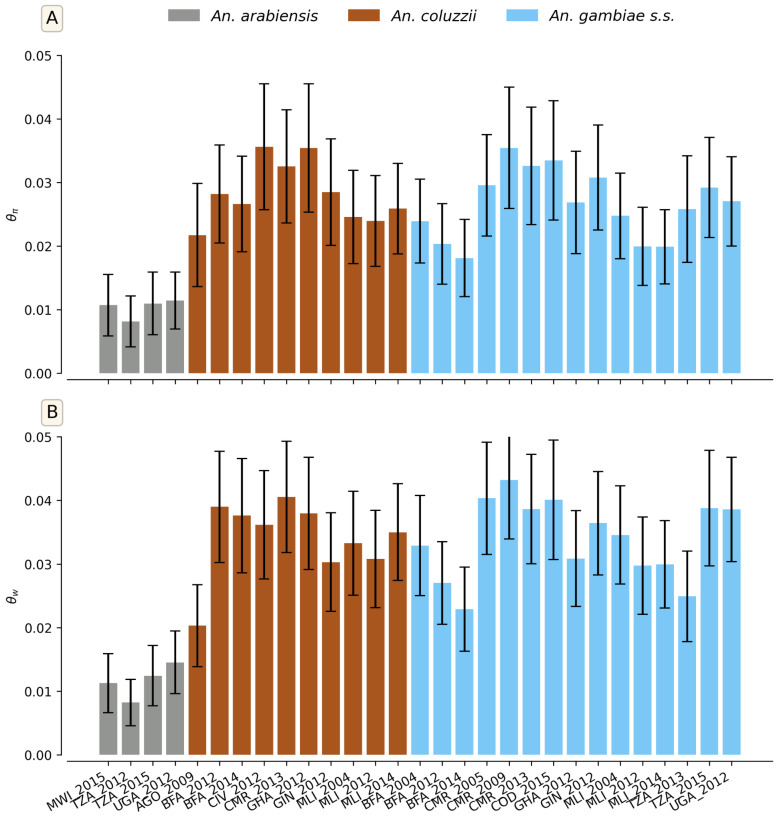
Genetic diversity statistics of *An. gambiae* s.l. populations at aldehyde oxidases locus. (**A**) Nucleotide diversity (θπ). (**B**) Watterson theta (θw). The X-axis shows the different cohorts (country/sampling period/species): AGO = Angola, BFA = Burkina Faso, CIV = Côte d’Ivoire, CMR = Cameroon, COD = Democratic Republic of Congo, GHA = Ghana, GIN = Guinea, MLI = Mali, MWI = Malawi, TZA = Tanzania, UGA = Uganda.

**Figure 8 genes-16-01018-f008:**
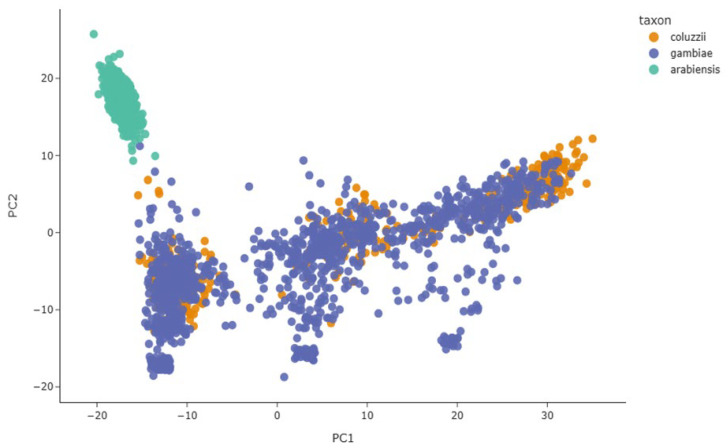
Principal component analysis of *An. gambiae* s.l. populations at aldehyde oxidases locus. X-axis shows the first principal component (PC1) and Y-axis the second principal component (PC2).

## Data Availability

Jupyter Notebooks and scripts to reproduce all the analyses, tables, and figures are available in the GitHub repository: https://github.com/HyacintheKi/aldehyde_data (accessed on 20 March 2025). The SNPs and haplotypes data are available on the homepage of MalariaGEN and can be accessed using the malariagen_data package. The raw sequences in FASTQ format and the aligned sequences in BAM format were stored in the European Nucleotide Archive (ENA, Study Accession n° PRJEB42254).

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
