# Peer review of "Identification of Novel Gene Cluster Potentially Associated with Insecticide Resistance in Anopheles gambiae s.l."

_genes, 2025, doi:10.3390/genes16091018_

Round 1
Reviewer 1 Report
Comments and Suggestions for Authors
The discussion is informative but contains overly long paragraphs and blends multiple topics (e.g., mechanisms of action, evolution of resistance, practical applications). It is recommended to restructure this section into clear subsections – roles of aldehyde oxidases, genetic mechanisms (SNPs vs. CNVs), geographic variations and gene flow, and implications for vector control.
The conclusion lacks a concise summary of specific numerical results (e.g., the number of high-frequency SNPs, geographic differences). Including concrete data would strengthen the overall impact and clarity of the findings.
Comments on the Quality of English LanguageThe language is scientifically accurate; however, there are noticeable repetitions and some unclear expressions (e.g., the frequent use of “our findings suggest”). A professional language editing by a native English speaker is recommended.
Author Response
Comments 1: The discussion is informative but contains overly long paragraphs and blends multiple topics (e.g., mechanisms of action, evolution of resistance, practical applications). It is recommended to restructure this section into clear subsections – roles of aldehyde oxidases, genetic mechanisms (SNPs vs. CNVs), geographic variations and gene flow, and implications for vector control.
Response 1: Thank you for this suggestion. We agree. Therefore, we have restructured the section into the following subsections: Aldehyde oxidases under positive selection, Potential role of aldehyde oxidases in insecticide resistance, Genetic mechanisms: SNPs and CNVs, Gene flow and geographic variation and Implications for vector control.
Comments 2: The conclusion lacks a concise summary of specific numerical results (e.g., the number of high-frequency SNPs, geographic differences). Including concrete data would strengthen the overall impact and clarity of the findings.
Response 2: Agree. We have accordingly incorporated specific results, including the frequencies of SNPs and CNVs, as well as a sentence summarizing the population structure and gene flow findings.
Comments 3: The language is scientifically accurate; however, there are noticeable repetitions and some unclear expressions (e.g., the frequent use of “our findings suggest”). A professional language editing by a native English speaker is recommended.
Response 3: Thank you for this valuable comment. We have carefully revised the manuscript to address repetitions and clarify ambiguous expressions. To improve the overall language quality, we sought assistance from colleagues proficient in scientific English and thoroughly edited the text. We believe these changes have enhanced the clarity and readability of the manuscript.
Reviewer 2 Report
Comments and Suggestions for Authors
Ki et al. address a critical challenge in malaria vector control—the rise of insecticide resistance—and emphasizes the need for genomic surveillance, which is well articulated. The use of high-quality whole genome sequencing data enhances the reliability of the results. Identification of both haplotype patterns and various types of genetic variation (SNPs, CNVs) provides a comprehensive view of potential resistance mechanisms. The mention of gene flow between An. gambiae and An. coluzzii adds valuable evolutionary and ecological context.
Author Response
Comments: Ki et al. address a critical challenge in malaria vector control—the rise of insecticide resistance—and emphasizes the need for genomic surveillance, which is well articulated. The use of high-quality whole genome sequencing data enhances the reliability of the results. Identification of both haplotype patterns and various types of genetic variation (SNPs, CNVs) provides a comprehensive view of potential resistance mechanisms. The mention of gene flow between An. gambiae and An. coluzzii adds valuable evolutionary and ecological context.
Response: We sincerely thank you for your positive and encouraging feedback. We truly appreciate your recognition of the clarity and significance of our work.
Reviewer 3 Report
Comments and Suggestions for Authors
The manuscript "Identification of novel gene clusters potentially associated with insecticide resistance in Anopheles gambiae s.l." is a very interesting study that reveals a potential participation of the aldehyde oxidase cluster in insecticide resistance. The authors found this interesting correlation by analyzing genomes of Anopheles gambiae complex mosquitoes. The experimental strategy is highly innovative, and the results are well discussed.
I just have some minor comments:
-Figures should be cited alongside the text whenever a result is described. For example, in line 144, the paragraph discusses a result, but it is not clear if it refers to Figure 2. It is essential to cite Figure 2. The authors should be consistent in citing figures throughout the text when describing results.
-Figures 5 and 6 say: Identical haplotypes are represented by nodes (circles). The larger the node, the greater the number of identical haplotypes. This sentence is true; however, it would be much better if the authors include and scale with circles of different sizes to give an idea of the number of identical haplotypes in each node.
-Figure 8 is cited before Figure 7; it would be better if you swap the order to have a better flow in the text.
Author Response
Comments 1: Figures should be cited alongside the text whenever a result is described. For example, in line 144, the paragraph discusses a result, but it is not clear if it refers to Figure 2. It is essential to cite Figure 2. The authors should be consistent in citing figures throughout the text when describing results.
Response 1: Thank you for pointing this out. We agree with this comment. Therefore, we have carefully revised the manuscript to ensure that all figures are cited consistently in the text when describing corresponding results. Specifically, Figure 2 has now been cited at line 138 (previously line 144).
Comments 2: Figures 5 and 6 say: Identical haplotypes are represented by nodes (circles). The larger the node, the greater the number of identical haplotypes. This sentence is true; however, it would be much better if the authors include and scale with circles of different sizes to give an idea of the number of identical haplotypes in each node.
Response 2: Thank you for this suggestion. As recommended, Figures 5 and 6 have been updated to include a scale with circles of different sizes, providing a visual indication of the number of identical haplotypes in each node.
Comments 3: Figure 8 is cited before Figure 7; it would be better if you swap the order to have a better flow in the text.
Response 3: Thank you for pointing that out. We have swapped the order of Figures 7 (line 212) and 8 (line 215) in the text to ensure a more logical flow and consistent sequence of figure citations.